# An H-GrabCut Image Segmentation Algorithm for Indoor Pedestrian Background Removal

**DOI:** 10.3390/s23187937

**Published:** 2023-09-16

**Authors:** Xuchao Huang, Shigang Wang, Xueshan Gao, Dingji Luo, Weiye Xu, Huiqing Pang, Ming Zhou

**Affiliations:** 1School of Automation, Guangxi University of Science and Technology, Liuzhou 545000, China; 221068359@stdmail.gxust.edu.cn (X.H.);; 2Key Laboratory of Intelligent Sensing and Control, Liuzhou 545000, China; 3Mechanical and Electrical College, Beijing Institute of Technology, Beijing 100190, China; 4Hangke Jinggong Co., Ltd., Beijing 102400, China

**Keywords:** indoor pedestrian segmentation, image enhancement, H-GrabCut algorithm

## Abstract

In the context of predicting pedestrian trajectories for indoor mobile robots, it is crucial to accurately measure the distance between indoor pedestrians and robots. This study aims to address this requirement by extracting pedestrians as regions of interest and mitigating issues related to inaccurate depth camera distance measurements and illumination conditions. To tackle these challenges, we focus on an improved version of the H-GrabCut image segmentation algorithm, which involves four steps for segmenting indoor pedestrians. Firstly, we leverage the YOLO-V5 object recognition algorithm to construct detection nodes. Next, we propose an enhanced BIL-MSRCR algorithm to enhance the edge details of pedestrians. Finally, we optimize the clustering features of the GrabCut algorithm by incorporating two-dimensional entropy, UV component distance, and LBP texture feature values. The experimental results demonstrate that our algorithm achieves a segmentation accuracy of 97.13% in both the INRIA dataset and real-world tests, outperforming alternative methods in terms of sensitivity, missegmentation rate, and intersection-over-union metrics. These experiments confirm the feasibility and practicality of our approach. The aforementioned findings will be utilized in the preliminary processing of indoor mobile robot pedestrian trajectory prediction and enable path planning based on the predicted results.

## 1. Introduction

In recent years, rapid advancements in robot technology have led to its extensive application in various domains such as goods delivery, elderly care, environment cleaning, and blind navigation. In indoor environments, the presence of numerous dynamic objects, predominantly pedestrians, adversely affects self-localization and path planning. Additionally, uneven indoor lighting conditions result in blurred pedestrian images with low brightness, making it difficult to identify distinct feature points and capture the background along with pedestrians. As a consequence, obtaining accurate distance measurements between pedestrians becomes challenging, which hampers obstacle avoidance and precise trajectory prediction for robots. Extracting pedestrians as regions of interest can eliminate irrelevant background information behind them and enable the accurate retrieval of their position coordinates in RGB images, thus facilitating the calculation of distances between pedestrians and depth cameras. This information is vital for pedestrian trajectory recovery or prediction. Therefore, the recognition and segmentation of dynamic obstacles, specifically pedestrians, from images are crucial in enabling robots to respond promptly to such obstacles. The research conducted in this study holds significant importance in the field of robot obstacle avoidance.

To address the issues of unstable noise and indoor illumination in the segmentation algorithm, the implementation of image enhancement can be considered. While traditional histogram equalization technology may enhance image brightness and contrast, it is likely to cause over-enhancement and lacking robustness. To overcome these challenges, scholars have proposed various image enhancement algorithms. For instance, the adaptive histogram equalization method [1,2,3] and the image difference algorithm [4,5] can effectively avoid unnatural contrast and distortion effects. Nonetheless, current methods are inadequate in improving contrast while retaining the integrity of image boundary features. Various researchers have proposed a Retinex enhancement algorithm based on image fusion and illumination image estimation [6,7,8,9,10]. This algorithm is capable of producing high-quality images, but the contour edges are typically not complete. In conclusion, there is currently no single optimal method to address the issues of contrast enhancement and the preservation of image boundary features simultaneously.

In order to extract distinctive feature points, depth cameras are used to segment pedestrians in the input images as regions of interest. Currently, both deep learning-based and traditional interactive algorithms are commonly employed for image segmentation. Examples of these methods include U-Net segmentation [11], the multi-scale high-order attention algorithm [12], DeepLabv3+ image segmentation [13], the motion object segmentation algorithm [14], and the local alignment network model [15]. While deep learning-based segmentation algorithms can achieve accurate segmentation of target objects, they also increase the complexity of the algorithms and lead to the excessive and redundant use of computational resources when extracting low-level features repeatedly. In addition, traditional interactive image segmentation algorithms have been a focus of research by many scholars, for instance, the GraphCut algorithm [16] and the GrabCut algorithm [17], where the GrabCut algorithm is simple, efficient, and provides high segmentation accuracy. There are also improved algorithms that combine the local sampling of the foreground and background [18], the adaptive *k*-means clustering algorithm [19], and the utilization of YOLO-V4 for foreground detection [20,21]. These algorithms have demonstrated a certain reference value in practical applications. However, the aforementioned segmentation algorithms are only suitable for the background segmentation of non-target pedestrians and are not applicable for avoiding dynamic obstacles. Indoor environments vary in complexity, lighting conditions change frequently, and images often contain noise. When moving pedestrians are close to the camera, their imaging brightness is low and clothing edge information tends to be blurred and easily fused.

This paper proposes an improved image segmentation algorithm called H-GrabCut to address the aforementioned issues and achieve background removal in complex pedestrian backgrounds. The H-GrabCut (Human-GrabCut) algorithm presented in this paper enhances the GrabCut algorithm’s clustering features by incorporating the BIL-MSRCR (Bilateral-MSRCR)image enhancement algorithm and introduces two-dimensional information entropy, UV component distance, and LBP texture features. This optimization effectively resolves the resource limitations associated with nesting multiple deep learning algorithms, while enabling more accurate distance measurement between robots and pedestrians, beyond relying solely on the closest distance for ranging. Furthermore, the algorithm demonstrates excellent performance in experiments, offering a new option for precise indoor pedestrian image segmentation in mobile robotics.

## 2. Method Construction

In indoor environments, the presence of numerous dynamic objects, primarily pedestrians, has a detrimental impact on the self-localization and path planning of indoor navigation robots. Additionally, uneven indoor lighting conditions result in pedestrian images captured by the visual system being characterized by low brightness, blurred edges, unclear contours, and prominent noise. These factors degrade the robustness of the traditional GrabCut segmentation algorithm, rendering feature points indistinct and possibly including background information, thereby impeding accurate distance estimation between pedestrians and introducing errors that affect the precision of obstacle avoidance and pedestrian trajectory prediction by the robot. To address the aforementioned issues, an improved algorithm called H-GrabCut (Human-GrabCut) is proposed. The algorithm consists of the following steps, as illustrated in Figure 1: (1) the preliminary noise reduction of the input image using range adaptive fast median filtering; (2) initial segmentation region partitioning through pedestrian detection using YOLO-V5; (3) the extraction of cropped images for each pedestrian bounding box, generating an image sequence for further processing; (4) the application of the improved BIL-MSRCR (Bilateral-MSRCR) algorithm to the pedestrian images in the sequence; and (5) the segmentation of the processed images using the H-GrabCut algorithm to obtain the final segmented image with pedestrian RGB coordinates.

### 2.1. Range Adaptive Fast Median Filtering

As shown in Figure 2, the depth camera based on binocular structured light 3D sensor technology mainly consists of left and right infrared cameras, a laser projection module, and a depth computing processor. The laser projection module is used to project the structured light pattern (speckle pattern) to the target scene, and the left infrared camera and the right infrared camera collect the left infrared structured light image and the right infrared structured light image of the target respectively. After receiving the left infrared structured light image and the right infrared structured light image, the depth calculation processor executes the depth calculation algorithm and outputs the depth image of the target scene.
(1)$$Z=\frac{f\times B}{x_{1}-x_{2}}$$

In Formula (Equation 1), *f* represents the focal length of the camera imaging plane; *B* denotes the baseline between the left and right infrared cameras; and $x_{1}-x_{2}$ signifies the disparity value of each pixel.

In this paper, the distance between the pedestrian and the robot was measured using a depth camera. When the distance exceeded one meter, a fast median filter was used for background filtering. The fast median filter selects the middle value of pixels in the filter window and counts pixels using a histogram. It then compares the pixels moving out of the window with the pixels moving in. The most effective filter results were achieved using a 3 × 3 window.

### 2.2. YOLO-V5 Pedestrian Target Detection

Since this algorithm focuses only on optimizing pedestrian segmentation, it detects pedestrians in the filtered images and assigns anchor boxes to them, with these anchor box points representing the foreground regions. The YOLO-V5 object detection network architecture consists of a backbone network and a detection head. The backbone network utilizes the CSP (cross stage partial) architecture, which includes multiple CSP blocks and an SPP (spatial pyramid pooling) module [21]. The detection head comprises three convolutional layers used for predicting the object’s category and bounding box. The overall image and coordinate information data transmission structure is illustrated in Figure 3.

As our algorithm focuses on indoor pedestrian segmentation, we utilize the YOLO-V5 framework for pedestrian recognition. However, we modify its detection head to output only one pedestrian label and intercept the image in the anchor frame to generate the required image sequence output. This procedure allows us to obtain the necessary foreground target for the pedestrian segmentation algorithm, thereby minimizing user operations. Additionally, we optimize the algorithm to handle numerous images quickly while maintaining high recognition accuracy and stability.

### 2.3. MSRCR Image Enhancement Algorithm Combined with Bilateral Filtering

In indoor environments, the image captured by a depth camera is often noisy due to poor lighting, and when pedestrians move in close proximity to the camera, the overall image brightness is low. Moreover, the edge information of pedestrians wearing different clothes is unclear, and similar backgrounds tend to fuse. Thus, image enhancement is necessary. The MSRCR algorithm [22], based on Retinex theory, aims to improve image brightness, contrast, and color balance. The formula is as follows:(2)$$\log R_{i}\left(x,y\right)=\sum_{k=1}^{N}\omega_{k}\left(\log\left(I_{i}\left(x,y\right)\right)-\log\left(I_{i}\left(x,y\right)\times G_{k}\left(x,y\right)\right)\right)$$

In Formula (Equation 2), k≤N and *N* represent the number of scales. Generally speaking, when *N* is 3, Gaussian filters are used to filter the original image on three different scales, respectively. $\omega_{k}$ represents a weight value for each scale, which must satisfy:(3)$$\sum_{k=1}^{N}\omega_{k}=1$$

$G_{k}$ is a Gaussian function with a number of scales *k*, and its formula is as follows:(4)$$G_{k}\left(x,y\right)=\frac{1}{\sqrt{2\pi }c_{k}}\times \exp\left[-\frac{x^{2}+y^{2}}{2c_{k}^{2}}\right]$$

In Formula (Equation 4), $x^{2}$ and $y^{2}$ are, respectively, expressed as the distance between the remaining pixels in the neighborhood and the center pixel in the neighborhood, and $c_{k}$ is the standard deviation of the filter to control the amount of retained spatial details. However, $c_{k}$ values cannot be modeled and determined theoretically and are basically a balance between enhancing local dynamics (such as showing details in shadows) and color rendering.

The conventional MSRCR algorithm uses a Gaussian filter that employs a Gaussian weighted average of pixels in the vicinity of the center point. Such a filter takes into account only the spatial relationship between pixels but fails to consider pixel similarity. Consequently, this approach blurs the boundaries, including those of anchor points returned by the object detection algorithm, leading to a loss of features. On the other hand, the bilateral filter, a nonlinear filter, can effectively remove noise while maintaining boundary clarity [23]. Bilateral filtering uses both spatial Gaussian weights and color-similarity Gaussian weights simultaneously. The spatial Gaussian function ensures that only pixels in the neighboring area affect the center point, whereas the color-similarity Gaussian function ensures that only pixels with similar values to the center pixels are considered for fuzzy operations. Thus, this study proposes the BIL-MSRCR algorithm, an improved bilateral filtering method applied to the MSRCR algorithm. As a result, the MSRCR algorithm can better preserve the image’s edge and texture structure.

The spatial domain of the improved bilateral filtering algorithm is expressed as:(5)$$h\left(x_{c}\right)=k_{d}^{-1}\;\int \int \;f\left(r\right)s\left(r,x_{c}\right)\;dr$$
(6)$$k_{d}^{}\left(x_{c}\right)=\;\int \int \;s(r,x_{c})\;dr$$

Formulas (Equation 5) and (Equation 6), $s(r,x_{c})$, represent the spatial distance function between the central pixel $x_{c}$ and its neighboring pixels *r*. The color domain in the two-sided filtering algorithm is represented as:(7)$$h\left(x_{c}\right)=k_{r}^{-1}\;\int \int \;f\left(r\right)c(f\left(r\right),f\left(x_{c}\right))\;dr$$
(8)$$k_{r}^{}\left(x_{c}\right)=\;\int \int \;c(f\left(r\right),f\left(x_{c}\right)\;dr$$

Formulas (Equation 7) and (Equation 8) represent the similarity between the central pixel value and the pixel value of its neighboring sample. In the above two filtering methods, f(r) represents the pixel value at *r* and $h(x_{c})$ represents the pixel value after filtering at $x_{c}$. Combining the above two filtering algorithm ideas, a new bilateral filter is generated, so the bilateral filter can be expressed as:(9)$$h\left(x_{c}\right)=k_{}^{-1}\;\int \int \;f\left(r\right)s\left(r,x_{c}\right)c\left(f\left(r\right),f\left(x_{c}\right)\right)\;dr$$
(10)$$k\left(x_{c}\right)=\;\int \int \;s\left(r,x_{c}\right)c(f\left(r\right),f\left(x_{c}\right))\;dr$$

In this paper, the spatial domain function takes the Gaussian filter as an example and is defined as:(11)$$s\left(r,x_{c}\right)=\exp\left(-\frac{{\left(r_{x}-x_{c}\right)}^{2}+{\left(r_{y}-y_{c}\right)}^{2}}{2c_{s}^{2}}\right)$$

In Formula (Equation 11), $\left(r_{x},r_{y}\right)$ represents the coordinate value of the current pixel, $\left(x_{c},y_{c}\right)$ represents the coordinate value of the center point of the image, and $c_{s}$ is the standard deviation parameter of the spatial domain.

The color domain function $c(f\left(r\right),f\left(x_{c}\right))$ is defined as:(12)$$c\left(f\left(r\right),f\left(x_{c}\right)\right)=\exp\left(-\frac{{\left(f\left(r_{x},r_{y}\right)-f\left(x_{c},y_{c}\right)\right)}^{2}}{2c_{c}^{2}}\right)$$

In Formula (Equation 12), $f(r_{x},r_{y})$ represents the pixel value of the current coordinate, $f(x_{c},y_{c})$ represents the pixel value of the center point of the image, and $c_{c}^{}$ is the standard deviation parameter of the color domain.

The global kernel function of bilateral filtering can be written from the above formula:(13)$$B_{k}\left(r_{x},r_{y}\right)=\exp\left[-\left(\frac{{\left(r_{x}-x_{c}\right)}^{2}+{\left(r_{y}-y_{c}\right)}^{2}}{2c_{s}^{2}}+\frac{{\left(f\left(r_{x},r_{y}\right)-f\left(x_{c},y_{c}\right)\right)}^{2}}{2c_{k}^{2}}\right)\right]$$

Although the kernel function-based bilateral filter in Formula (Equation 13) can preserve the edge structure information while denoising, its algorithm operates at a pixel-level. Consequently, the structure preservation effect is not significantly enhanced, and noise will compromise the calculation of the weighting coefficient, resulting in a reduced noise suppression ability. Additionally, the bilateral filter kernel function disregards the image’s features and spatial neighborhood. The spatial domain kernel and the color domain kernel in the kernel function receive the same weight, making the filter unsuitable for robust applications. To better preserve the image’s edge and detail structure, we assign more weight to the spatial domain kernel. We introduce a coefficient “*b*”, which adjusts between the spatial domain kernel and the color domain kernel.The overall kernel function of improved bilateral filtering is as follows.
(14)$$B_{k}=\exp\left[-\left(b\times \left(\frac{{\left(r_{x}-x_{c}\right)}^{2}+{\left(r_{y}-y_{c}\right)}^{2}}{2c_{s}^{2}}\right)+\left(1-b\right)\times \frac{{\left(f\left(r_{x},r_{y}\right)-f\left(x_{c},y_{c}\right)\right)}^{2}}{2c_{k}^{2}}\right)\right]$$

In Formula (Equation 14), *b* is used to adjust the coefficients of spatial domain kernel and color domain kernel, $x_{c}$ and $y_{c}$ are the coordinate values of the image center, $f(x_{c},y_{c})$ is the pixel value of the image center point, $c_{s}$ is the standard deviation of Gaussian function in space domain, and $c_{k}$ is the standard deviation of the Gaussian function in the color domain.

Based on the above improved bilateral filtering, the addition of the weighting coefficient enables the bilateral filtering to better balance the tradeoff between smoothing and preserving details, thus better preserving edge and texture information. Therefore, the overall formula of the BIL-MSRCR algorithm is as follows.
(15)$$\log R_{i}\left(x,y\right)=\sum_{k=1}^{N}\omega_{k}\left(\log\left(I_{i}\left(x,y\right)\right)-\log\left(I_{i}\left(x,y\right)\times B_{k}\left(x,y\right)\right)\right)k\leq N$$

In Formula (Equation 15), $R_{i}\left(x,y\right)$ is the pixel output of the *i*th color channel, $I_{i}\left(x,y\right)$ is the pixel input of the *i*th color channel of the BIL-MSRCR algorithm, and $B_{k}$ is the improved bilateral filter kernel function with scale number *k*.

This paper adopts an improved bilateral filter as the center surround function of Multi-Scale Retinex (MSRCR) for image decomposition and enhancement, allowing for a separation of the light image from its reflection component. The bilateral filter radius, specified in three distinct scales, is set to 5, 10, and 15. The results show that the optimal effect is achieved when *b* is 0.31 following experimental testing and verification. Subsequent image enhancements significantly enhance the image contrast while prohibiting boundary blurring and maintaining the image’s boundary features in its totality.

### 2.4. GrabCut Algorithm Combining Spatial Information, Chromaticity Information, and Texture Information

Iteratively, the GrabCut algorithm enhances the GMM parameters for modeling the target and background. The Gibbs energy function is then defined, and the algorithm segments the image by solving the energy function’s min-cut. The energy function is expressed below:(16)$$E\left(\alpha ,k,\theta ,p\right)=U\left(\alpha ,k,\theta ,p\right)+V\left(\alpha ,p\right)$$
(17)$$\theta =\left\{\pi (\alpha ,k),\mu (\alpha ,k),\Sigma (\alpha ,k),\alpha =(0,1),k=1\cdots K\right\}$$

In Formula (Equation 16), α represents the transparency coefficient, 0 is the background, and 1 is the foreground; Formula (Equation 17) represents the proportion corresponding to each GMM component; *p* represents a single pixel; Function *U* represents a regional data item of the energy function; and the function *V* represents the smooth term of the energy function.

In order to use the GrabCut algorithm [20], the user is required to mark the area of interest using a rectangular bounding box, whilst the image outside is regarded as the background and the pixels within the possible foreground and background. Employing the *k*-means algorithm, the pixels located inside the bounding rectangle are clustered into a *k* class system, which initializes *k* models of the Gaussian mixture model (GMM). Equation (Equation 18) is then used to calculate the RGB value of the pixels within the bounding box to determine its Gaussian component membership.
(18)$$k_{n}:\arg\min_{k_{n}}\;D\left(\alpha_{n},k_{n},\theta ,p_{n}\right)$$

In Equation (Equation 18), $\theta =(\mu ,\sigma^{2})$, in which μ is the mean value and $\sigma^{2}$ is the variance.

The region data item represents the negative logarithm of the probability that a pixel belongs to the background or foreground, which is calculated according to Equation (Equation 19):(19)$$D\left(\alpha_{n},k_{n},\theta ,p_{n}\right)=-\log\pi \times W_{n}+\frac{1}{2}\log\left(\det\Sigma W_{n}\right)+\frac{1}{2}\times Q_{n}^{T}\Sigma W_{n}^{-1}Q_{n}^{}$$

In Formula (Equation 19), $W_{n}=(\alpha_{n},k_{n})$ and $Q_{n}^{}=\left[p_{n}-\mu (\alpha_{n},k_{n})\right]$.

After determining the probability that each pixel belongs to the foreground and background, the smooth term is calculated according to Equation (Equation 20):(20)$$V\left(\alpha ,p\right)=\gamma \times \underset{(m,n)\in C}{\Sigma }\;dis{\left(m,n\right)}^{-1}\times \left[\alpha_{n}\neq \alpha_{m}\right]\times \exp\left[-\beta {\left(p_{m}-p_{n}\right)}^{2}\right]$$

In Formula (Equation 20), (m,n) represents pixel coordinates; *C* represents a set of adjacent color pairs; $dis{\left(m,n\right)}^{-1}$ represents the Euclidean distance between adjacent pixels, where 1 is taken; and $\beta ={\left(2\langle {\left(p_{m}-p_{n}\right)}^{2}\rangle \right)}^{-1}$, $\langle \cdot \rangle$ represents the expectation of the image sample.

According to the Gibbs energy term, the s-t graph is constructed, and the image is divided by Equation (Equation 21) maximum flow/minimum cut:(21)$$\min_{\left\{\alpha_{n}:n\in T_{U}\right\}}\;\min_{k}\;E\left(\alpha ,k,\theta ,p\right)$$

In the original GrabCut algorithm, the *k*-means algorithm only clusters the color information of the image. The following formula is the original pixel feature:(22)$$z_{i}=\left[p_{R},p_{G},p_{B}\right]$$

Formula (Equation 22) $p_{R},p_{G},p_{B}$ represents the RGB component value of a pixel point $z_{i}$. This feature is single, and it is easy to cause the classification error of pixels in the initial classification of images. Considering the overall information of the image, this paper is proposed to integrate spatial information, chroma information, and texture feature information of the image during clustering to optimize the parameters of the GMM model. The following is the optimized features:(23)$$z_{i}=\left[p_{R},p_{G},p_{B},p_{H},p_{D},p_{L}\right]$$

In Formula (Equation 23), $p_{H}$ represents the two-dimensional information entropy of pixel $z_{i}$, $p_{D}$ represents the distance between the UV component of pixel $z_{i}$ and the clustering center, and $p_{L}$ represents the LBP texture feature value of pixel $z_{i}$.

To improve the accuracy of the clustering algorithm in classifying color information [24], this paper adopts conversion of the image into YUV color space to separate its brightness and chroma components. The Y value in YUV is then used as the gray value, computing its two-dimensional information entropy thereby providing the feature input for clustering. The two-dimensional information entropy of Y component of pixels in clustering features indicates the quantity of information in the image, which is expressed using probability distribution function. To depict the spatial attributes of pixel distribution, this paper adopts the feature quantity $\left(i,j\right)$, where *i* represents the gray value (0≤i≤255) of the pixel while *j* represents the gray level (0≤j≤255) of neighboring pixels.
(24)$$P_{ij}=\frac{f\left(i,j\right)}{N^{2}}$$

In Formula (Equation 24), $f\left(i,j\right)$ is the frequency of the occurrence of the feature binary $\left(i,j\right)$, and *N* is the scale of the image; the two-dimensional information entropy of the image is described as:(25)$$H=-\sum_{i=0}^{255}\sum_{j=0}^{255}p_{ij}\log_{2}p_{ij}$$

Image information entropy provides insight into the amount of information contained within an image. To overcome the issue of spatial information depletion caused by using a single feature, adding two-dimensional information entropy to the clustering features accounts for the compositional arrangement of adjacent pixels and the image’s spatial distribution characteristics. The traditional *k*-means clustering algorithm can cause the loss of details in an image as it considers only the gray component whilst ignoring chroma saturation. To address this limitation, this paper introduces a novel distance formula that characterizes the UV component of chroma and the chroma factor of the cluster center point. With this new distance formula, the *k*-means algorithm can accommodate different image types, thereby improving the algorithm’s adaptability.
(26)$$D=\sqrt{{\left(V_{C}-V_{N}\right)}^{2}+{\left(U_{C}-U_{N}\right)}^{2}}$$

In Formula (Equation 26), $\left(U_{C},V_{C}\right)$ represents the UV value of the cluster center, and $\left(U_{N},V_{N}\right)$ represents the UV value of the current pixel point. Taking the distance between the UV component and the cluster center into the clustering features, the difference between different images can be more accurately reflected, thus improving the robustness of the algorithm.

It is necessary to consider the texture information of an image, in addition to its spatial and color information. The texture feature of each pixel is dependent on its relationship with other pixels within its domain. Distinct objects have unique texture features, which facilitates the better classification of images. This study utilizes the local binary pattern (LBP) operator to describe the image’s texture features. LBP possesses rotation and gray invariance. LBP uses a 3 × 3 template as a unit and adopts the middle pixel as its threshold. A pixel point is marked as 1 if the pixel value of its eight surrounding fields is greater than the middle pixel; otherwise, it is marked as 0. The comparison of these values constructs an 8-bit binary number. The conversion of this binary number to decimal yields the LBP value of the middle pixel and reflects its inherent texture information. The LBP operator is effectively expressed in Equation (Equation 27):(27)$$LBP\left(p_{c}\right)=\sum_{i=0}^{8}2^{i}p_{i}p_{i}=\left\{\begin{array}{c} 0,p_{c}>p_{i}\\ 1,p_{c}<p_{i} \end{array}\right.,i=1,2,\ldots ,8.$$

### 2.5. Algorithms Run Pseudo-Code

Based on the above theory, we can utilize the improved H-GrabCut algorithm to effectively segment images. The specific implementation method of the algorithm can be expressed by the following pseudo-code.

Algorithm 1 introduces our proposed BIL-MSRCR and H-GrabCut algorithms, which adhere to the following principles:Apply a range adaptive fast median filter with a window size of 3 × 3 to the input raw image, resulting in the filtered image $I_{Q}$.(Lines 4 and 5) Pass the image $I_{Q}$ to a function that combines the pedestrian bounding box positions obtained from YOLO-V5. Crop the image $I_{Q}$ based on these bounding box positions, generating a pedestrian image $I_{H}$ that contains a small background area.(Lines 8–14) Apply the BIL-MSRCR algorithm to image $I_{H}$, enhancing it to obtain the enhanced pedestrian image $I_{BMSR}$ (Equations (Equation 2)–(Equation 15)).(Line 15) Perform initial *k*-means clustering on the image to obtain cluster centers. Calculate the distance between the UV components of each pixel and the chromaticity factors of the cluster centers.(Line 16) Convert the enhanced $I_{BMSR}$ image from the RGB color space to the YUV color space. Compute the two-dimensional information entropy specifically for the Y channel of the $I_{BMSR}$ image (Equation (Equation 25)).(Line 17) Incorporate pixel spatial information, chromaticity information, and texture features from the $I_{BMSR}$ image into the *k*-means clustering process. Initialize the parameters of the Gaussian mixture model (GMM) using LBP components based on defined background pixels, potential foreground pixels, and potential background pixels (Equation (Equation 27)).(Line 18) Utilize the max-low algorithm to determine the min-cut. Iteratively apply (Equations (Equation 18)–(Equation 23)) to segment the $I_{BMSR}$ image, resulting in a foreground image mask denoted as $I_{MASK}$.(Line 19) Combine the mask image $I_{MASK}$ with the filtered output image $I_{Q}$ to generate the segmented image $I_{HG}$.
**Algorithm 1** BIL-MSRCR and H-GrabCut**Input:** Camera_Image, *b* and The xy coordinates returned by YOLO-V5**Output:** Mask_Image and Image after segmentation  1: $x_{i}=\left[r_{x}^{1},r_{y}^{1},r_{x}^{2},r_{y}^{2}\right]$;  2: $w_{k}=\left[w_{1},w_{2},w_{3}\right]$;  3: **while** Camera_Image **and** $x_{i}$ is not empty **do**  4:    $I_{Q}$ = Range_filtering(Camera_Image);  5:    $I_{H}$ = Cut_Process($I_{Q}$, $r_{x}^{1}$, $r_{y}^{1}$, $r_{x}^{2}$, $r_{y}^{2}$);  6:    **for** *i* = 0 **to** scale **step** 1**:**  7:     $\log(I_{i})$ = Convert_log($I_{H}$);  8:     **for** *k* = 0 **to** *N* **step** 1**:**  9:     $B_{k}$ = BIL_Process($I_{H}$, $w_{k}$, b);10:     $\log(I_{i}\times B_{k})$ = Convert_log($I_{H}$, $B_{k}$);11:     $\log(R_{i})$ = Sub($w_{k}$, $\log(I_{i}\times B_{k})$, $\log(I_{i})$);12:     $I_{\mathrm{BMSR}}$ = Convert_Scale($\log(R_{i})$);13:     **end for**14:    **end for**15:    D = Extract_YUV($I_{BMSR}$);16:    H = Extract_entropy($I_{BMSR}$);17:    L = Extract_LBP($I_{BMSR}$);18:    $I_{\mathrm{MASK}}$ = HGrabcut_process($I_{BMSR}$, *D*, *H*, *L*);19:    $I_{\mathrm{HG}}$ = add($I_{Q}$, $I_{MASK}$);20: **end while**

## 3. Results

The primary algorithm was implemented using the C++ programming language on the ROS platform. The execution of the algorithm required the OpenCV 4.5.5 library, which was configured in the Clion v2020.3 integrated development environment. The computing device utilized in the experiment had the following specifications: a 64-bit Ubuntu operating system, 3.6 GHz CPU frequency, and 16.0 GB memory. For image recognition and segmentation, an RGB-D camera was necessary. The robot’s structure is shown in Figure 4, and the camera’s output resolution was set to 640 × 480 @ 30 fps. To ensure stable and non-shaky images, the robot was configured with a maximum movement speed of 0.1 m/s.

### 3.1. Experimental Analysis of Image Enhancement

The materials utilized in this experiment are partial images from the INRIA dataset and photographs captured by real machines, as illustrated in Figure 5. To prove the effectiveness and robustness of the proposed algorithm, we use the image enhancement algorithms proposed in the literature, such as CLAHE [3], GSMSR [6], LIME [7], RFVM [8], SRME [9], ALSM [10] as reference, the visual effect and gray-level histogram as indicators. Brightness, contrast, and naturalness in images can be perceived by the human vision system, while the histogram verifies the dynamic range and rationality of the pixel distribution.

The enhancement effects of various methods on the image Figure 5c are shown in Figure 6. The original image Figure 6a is dark with low contrast. It is evident that the enhanced image in Figure 6b exhibits significant image distortion and over-enhancement. The brightness enhancement in Figure 6d,e is insufficient as some details in dark areas are still not visible. The overall brightness enhancement in Figure 6c,g is still inadequate, and the distant scenery in dark areas remains unclear. The enhancement results in Figure 6f and our proposed method are better, with suitable overall brightness. However, upon careful observation, it can be noticed that Figure 6f exhibits over-enhancement in distant scenery with minor color distortion, whereas our method Figure 6h achieves appropriate overall enhancement, higher contrast, and better naturalness of the image.

A histogram provides an intuitive evaluation of image quality. An ideal image should meet two criteria. First, it should fully utilize the pixel-level space, ensuring high contrast and clear texture details. Second, pixels should be evenly distributed across the pixel-level space, resulting in appropriate brightness and natural colors. The histograms for the image enhancement algorithms in Figure 6a are displayed in Figure 7. Figure 7a shows that while the original image utilizes the entire pixel-level space, it mainly concentrates pixels at low gray levels, which darkens the image and obscures details. The histograms of the enhancement algorithms all utilize the full gray scale space, yielding good contrast. However, Figure 7b shows over-enhanced pixels with excessive color contrast. Figure 7c exhibits a relatively even pixel distribution, with slight unevenness in high gray scales. In Figure 7d and g, the pixel distribution peaks closer to higher gray scales, displaying uneven areas and obscuring details in darker regions. Figure 7e has a relatively even pixel distribution but leans slightly towards lower levels, resulting in an overall darker appearance. Both Figure 7f and our method offer even pixel distribution, suitable brightness, natural colors, and rich texture and details. However, Figure 7f exhibits over-enhanced pixels at specific levels.

### 3.2. Experimental Analysis of Segmentation Algorithm

We utilized the YOLO-V5 object detection algorithm, initially pretrained on the INRIA pedestrian dataset containing 614 images. To enhance algorithm performance in detecting pedestrians under diverse conditions, including occlusions, lighting variations, and various scenes, we expanded the dataset with an additional 100 real-world images. To validate our proposed target detection and segmentation algorithm, we tested it using both real-world images and the INRIA pedestrian dataset. YOLO-V5 was employed to identify foreground targets with a confidence threshold of 0.85 for automatic pedestrian segmentation. You can find the experimental results in Figure 8.

### 3.3. Contrast Experiment

We conducted experiments to assess the performance of our proposed algorithm for intricate background removal in comparison to existing algorithms. In our evaluations, we employed a dataset consisting of 300 pedestrian images, comprising 250 images sourced from the INRIA test dataset and an additional 50 real-world images. These images posed various challenges, encompassing diverse and complex backgrounds, occlusions, varying lighting conditions, and scene compositions. We compared the outcomes of our algorithm with those of the conventional GrabCut algorithm [17]. as well as algorithms proposed in prior literature LSGC [18], A-KGC [19]. Figure 9 illustrates a subset of the comparative results.

Figure 9 illustrates that the original GrabCut algorithm’s lack of an object recognition feature necessitates the manual division of foregrounds, resulting in superfluous background elements. Furthermore, it lacks image filtering and enhancement tools, contributing to its sensitivity to noise and low robustness to illumination. Consequently, indistinct pedestrian edges and undersegmentation often occur, as shown in Figure 9b,c. Although the algorithm proposed by reference [18] achieves desirable pedestrian background removal in simple scenes, it may fail to accurately segment the pedestrian area in complex backgrounds with uneven image gray levels, as shown in Figure 9c. In the case of complex indoor pedestrian backgrounds, local sampling becomes more sensitive to noise and outliers leading to over-segmentation. Conversely, the algorithm demonstrated in Figure 9d by reference [19] provides superior results in pedestrian background removal. In comparison to other segmentation algorithms, the proposed algorithm generates clear pedestrian contours with complete details, as shown in Figure 9e.

### 3.4. Segmentation Result Analysis

In terms of segmentation effect measurement, the following performance indicators were used to further evaluate our method [21]: sensitivity (*Sen*), error segmentation rate (*E*), intersection over union (*IOU*), accuracy (*Acc*), and relative performance improvement (*RPI*). All of the above assessment indicators are calculated by the following formula:(28)$$Acc=\frac{TP+TN}{TP+TN+TN+FP}$$
(29)$$Sen=\frac{TP}{TP+FN}$$
(30)$$IOU=\frac{TP}{TP+FN+FP}$$
(31)$$E=\frac{FP}{TP+FP+FN+TN}$$
(32)$$RPI=\frac{\left(Original\_Time-Improved\_Time\right)}{Original\_Time}\times 100\%$$
where *TP* (true example) represents the number of correctly segmented pixels. *FP* (false positive example) indicates the number of pixels separated by the error. *TN* (true negative example) is correctly unsegmented pixels. *FN* (false negative example) is the number of pixels that are not segmented by errors. The performance indicators of each image are shown in Figure 10, and the specific performance indicators are shown in Table 1:

Table 1 shows that the proposed H-GrabCut image segmentation algorithm outperforms other segmentation techniques in terms of sensitivity (*Sen*), error segmentation rate (*E*), intersection over union (*IOU*), accuracy (*Acc*), relative performance improvement (*RPI*), and average processing time. This outcome suggests that the proposed algorithm is efficient at eliminating noise. For images with complicated backgrounds, we implemented an enhanced image segmentation approach, which significantly reduced the segmentation error rate. The results of this approach indicate that post-segmentation pedestrian edges are smooth with clear delineation, further affirming the efficacy of the proposed algorithm.

## 4. Conclusions

Inspired by the distance between moving obstacles and robots in robot collision avoidance, this study proposes an improved H-GrabCut segmentation algorithm to remove the background of indoor pedestrians in the RGB space. Experimental research is conducted to introduce clustering features that are more effective than those of traditional algorithms. These features preserve the integrity of pedestrian image boundaries and effectively reduce the number of algorithm iterations, providing the optimal measurement point for pedestrian distance and improving the accuracy and reliability of image segmentation.

We conducted tests on the enhanced H-GrabCut algorithm using both real-world scenarios and the INRIA pedestrian dataset. We compared its segmentation results with those of the original GrabCut, local sampling GrabCut (LSGC), and adaptive *k*-Means GrabCut (A-KGC) algorithms. While all three algorithms showed similar performance in segmenting the central regions of pedestrians, a comprehensive comparison involving error segmentation rates, segmentation accuracy, and average intersection over union (IOU) revealed that H-GrabCut outperformed the others. The experiments confirmed the feasibility and practicality of our proposed H-GrabCut algorithm, providing convenience for indoor pedestrian trajectory prediction and obstacle avoidance research in our laboratory. We have developed a system for precise background segmentation on a mobile robot and plan to extend our research to indoor pedestrian trajectory prediction and obstacle avoidance, while considering the stability of background segmentation under varying robot speeds.

## Figures and Tables

**Figure 1 sensors-23-07937-f001:**
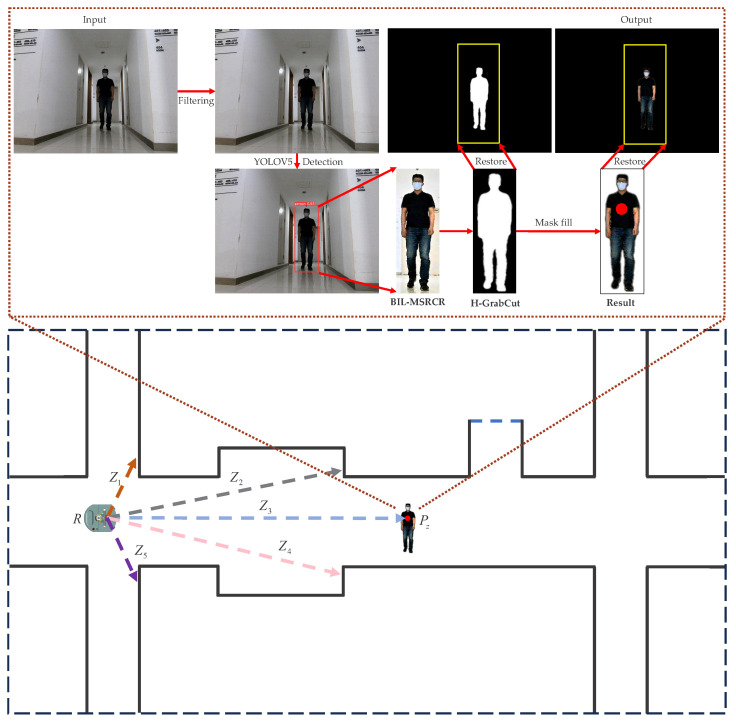
Flowchart of the H-GrabCut algorithm. (**Top**) The input is an RGB original image, and the output of the H-GrabCut algorithm is a background-removed image with pedestrian RGB coordinates. (**Bottom**) This image demonstrates the distance measurement of various obstacles (pedestrians) by the mobile robot *R* in an indoor environment using a depth camera. The distances between robot *R* and the walls are represented by $Z_{1}$, $Z_{2}$, $Z_{4}$, and $Z_{5}$ (orange, black, pink, and purple), while the $Z_{3}$ (blue) indicates the distance between robot *R* and the pedestrian $P_{z}$ (red).

**Figure 2 sensors-23-07937-f002:**
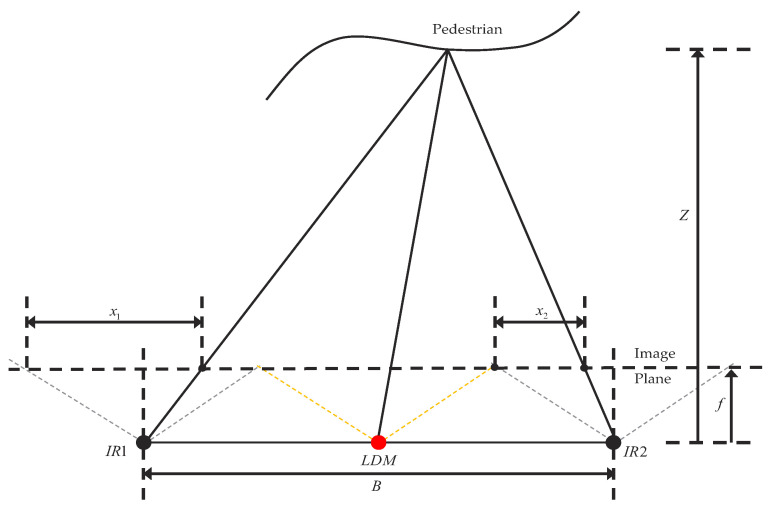
A schematic diagram of a binocular structured light 3D sensor is shown. In this diagram, IR1 and IR2 denote the infrared modules, and LDM stands for the laser module. The green line signifies the laser module’s maximum scanning angle, while the brown line indicates the maximum receiving angle of the infrared modules.

**Figure 3 sensors-23-07937-f003:**
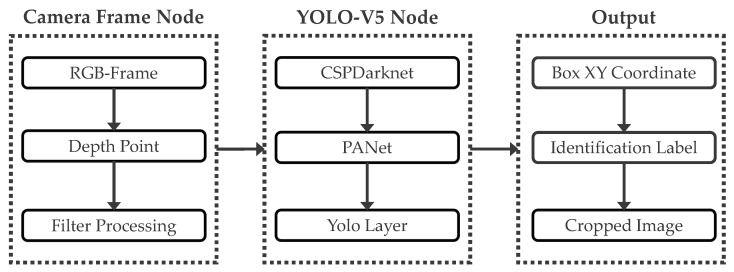
Structure of YOLO-V5 target recognition algorithm. The camera frame node is responsible for capturing the essential camera image data and applying filtering processes to produce an output image. This output image is subsequently fed into the YOLO detection layer for pedestrian annotation. Upon completion of annotation, the XY coordinates of the output anchor boxes can be obtained, along with the RGB images cropped based on these coordinates and their corresponding recognition labels.

**Figure 4 sensors-23-07937-f004:**
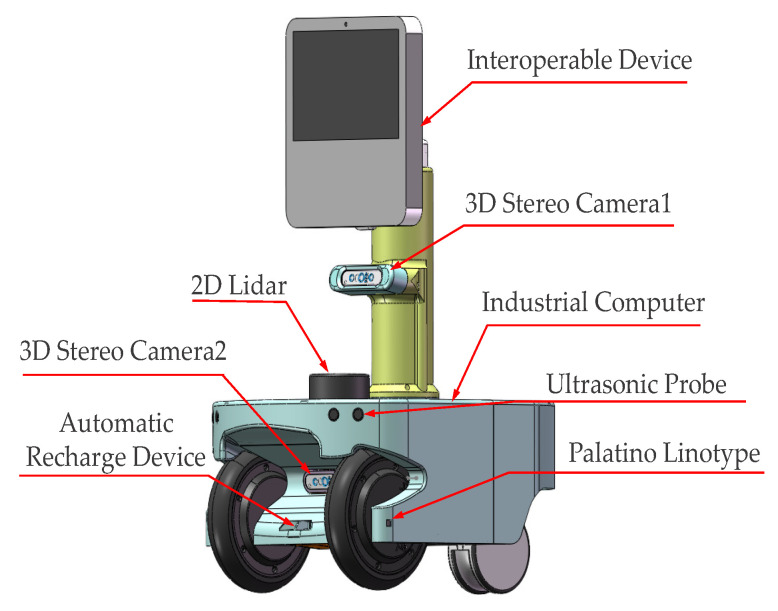
Robot platform sensor layout: interactive device is used to upload sensor data and receive motion control instructions, 3D stereo phase unit is used to return distance information and image information, and the industrial computer is used to run the ROS platform for control.

**Figure 5 sensors-23-07937-f005:**
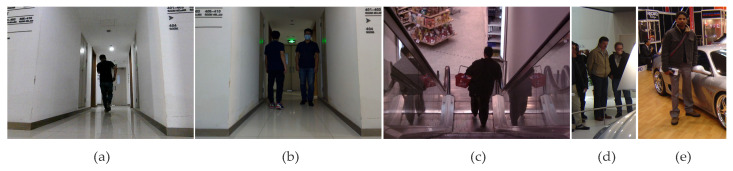
Real machine return images (**a**,**b**) and some images of INRIA pedestrian dataset (**c**–**e**).

**Figure 6 sensors-23-07937-f006:**
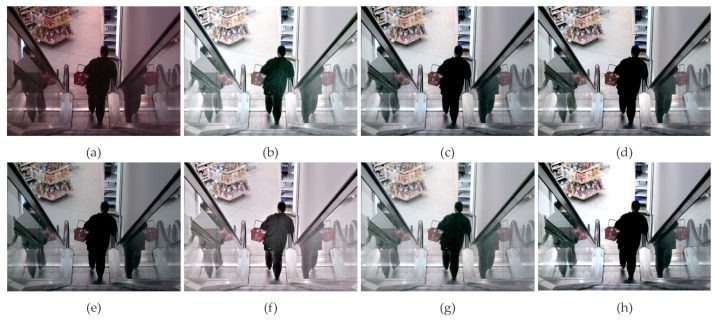
(**a**) is the RGB original of Figure 5c, (**b**) is the CLAHE transformation algorithm used in reference, (**c**) adopts the gray scale adaptive msr algorithm GSMSR, and (**d**) adopts the LIME algorithm. (**e**) adopts the improved RFVM algorithm, (**f**) is the SRME algorithm using the improved regular term, (**g**) is the ALSM algorithm, and (**h**) is the BIL-MSRCR algorithm proposed in this paper.

**Figure 7 sensors-23-07937-f007:**
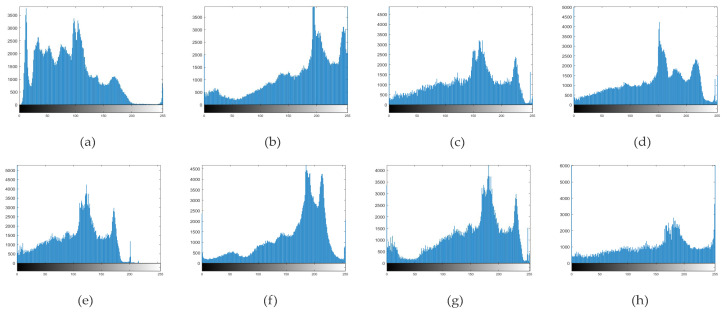
(**a**) is the gray scale histogram of the original Figure 5c, (**b**) is the gray scale histogram of the output image of the CLAHE algorithm, (**c**) is the gray scale histogram of the output of the GSMSR, (**d**) is the histogram of the LIME algorithm, (**e**) is the output histogram of the RFVM algorithm, and (**f**) is the histogram of the output of the SRME algorithm, (**g**) adopts the histogram effect output by ALSM, and (**h**) is the output histogram of the algorithm in this paper.

**Figure 8 sensors-23-07937-f008:**
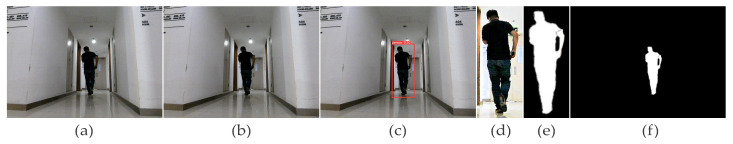
(**a**) is the RGB original image of Figure 5a, (**b**) the fast median filter is used to remove the salt and pepper noise caused by poor light conditions; YOLO-V5 target recognition algorithm was further used to identify pedestrians and reduce them according to the anchor frame size, as shown in (**c**). The BIL-MSRCR algorithm was used to improve the overall brightness of the cropped image, while preserving key details such as edges, as shown in (**d**). Finally, H-GrabCut algorithm was used to segment the pedestrian image with the background removed, as shown in (**e**). At last, the mask generated by H-GrabCut was covered back to the original image, as shown in (**f**).

**Figure 9 sensors-23-07937-f009:**
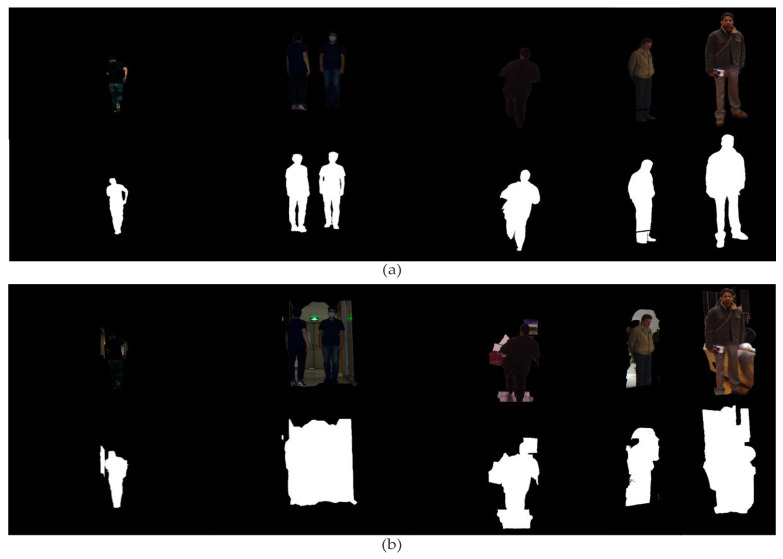
(**a**) portrays the pedestrian standard segmentation in Figure 5, whereas (**b**) shows the pedestrian segmentation acquired via the original GrabCut algorithm. The results of the improved GrabCut algorithm utilizing local sampling LSGC and adaptive *k*-means A-KGC can be seen in (**c**,**d**), respectively. Lastly, in (**e**), the segmentation results of the algorithm presented in this paper are depicted.

**Figure 10 sensors-23-07937-f010:**
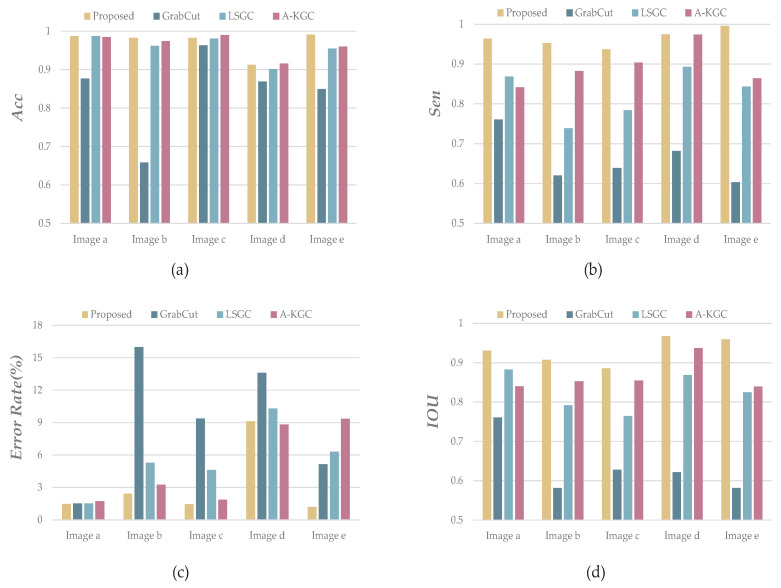
(**a**) is the *Acc* score of each image in Figure 5 in different segmentation algorithms; (**b**) Figure is the *Sen* score bar chart of these 5 images; (**c**) is the binary Error segmentation rate, expressed as a percentage; and (**d**) is the *IOU* score in 4 different segmentation algorithms.

**Table 1 sensors-23-07937-t001:** Performance index data of different segmentation algorithms.

Method	ACC (%)	SEN (%)	IOU (%)	E (%)	RPI (%)	Time (s)
Grab Cut	84.334	66.102	63.462	8.7650	0.00	6.35
LSGC [18]	95.728	82.563	82.647	5.8062	20.78	5.03
A-KGC [19]	96.482	89.307	86.457	5.0058	50.23	3.16
Proposed	97.133	96.481	93.032	3.1277	69.29	1.95

## Data Availability

Not applicable.

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
