# Peer review of "An H-GrabCut Image Segmentation Algorithm for Indoor Pedestrian Background Removal"

_sensors, 2023, doi:10.3390/s23187937_

Round 1

Reviewer 1 Report

Avery apt problem definition in today's scenario.

Well organized writeup.

Author Response

We are truly grateful for your vote of confidence in our manuscript. Your encouragement reinforces our commitment to producing high-quality research. Thank you for your valuable support and consideration.

Reviewer 2 Report

The paper is focused on the field of computer vision, specifically on segmentation algorithms for background removal in indoor applications. Authors have improved H-GrabCut algorithm and compared with known ones. Methodology of the paper is clear and correct and the work described in the paper is promising.

Figure 9 including letter-labels is unnecessarily large.

In algorithm description, the “Equation 2-15” notation is used but there is not such equation in the text.

The significance of the research is mainly intended for moving indoor robots, but the change of movement or velocity is not included in the experiments.

The proposed approach was tested on own and the INRIA dataset. However, it is not mentioned how large the dataset is, nor whether multiple pedestrians in one shot were considered for the own data.

When evaluating time complexity, it is more appropriate to use relative time changes related to, for example, the original algorithm. The specific hardware does not matter.

Some identical statements are repeated in several places in the text. In many places, spaces for punctuation and font style for symbols are not used correctly.

Author Response

We would like to express our sincere gratitude for your diligent work and valuable insights. Your constructive comments and suggestions have significantly contributed to the enhancement of our manuscript. We deeply appreciate your time and effort in providing a thorough review of our work, which has helped us refine our research and present it more effectively. Your commitment to the peer review process is highly commendable, and we are thankful for your support in advancing the quality of our paper.

Reviewer 3 Report

Article is well articulated and the topic discussed is novel.

Literature survey and experimental results are convincing.

I consider the article can be accepted in its present form.

Author Response

Thank you for your positive assessment, which motivates us to continue our research endeavors.
